# Tumor Microenvironment and Metabolism: Role of the Mitochondrial Melatonergic Pathway in Determining Intercellular Interactions in a New Dynamic Homeostasis

**DOI:** 10.3390/ijms24010311

**Published:** 2022-12-24

**Authors:** George Anderson

**Affiliations:** CRC Scotland & London, Eccleston Square, London SW1V 1PG, UK; anderson.george@rocketmail.com

**Keywords:** tumor microenvironment, mitochondria, melatonin, N-acetylserotonin, aryl hydrocarbon receptor, kynurenine, gut microbiome, circadian, treatment

## Abstract

There is a growing interest in the role of alterations in mitochondrial metabolism in the pathoetiology and pathophysiology of cancers, including within the array of diverse cells that can form a given tumor microenvironment. The ‘exhaustion’ in natural killer cells and CD8+ t cells as well as the tolerogenic nature of dendritic cells in the tumor microenvironment seems determined by variations in mitochondrial function. Recent work has highlighted the important role played by the melatonergic pathway in optimizing mitochondrial function, limiting ROS production, endogenous antioxidants upregulation and consequent impacts of mitochondrial ROS on ROS-dependent microRNAs, thereby impacting on patterned gene expression. Within the tumor microenvironment, the tumor, in a quest for survival, seeks to ‘dominate’ the dynamic intercellular interactions by limiting the capacity of cells to optimally function, via the regulation of their mitochondrial melatonergic pathway. One aspect of this is the tumor’s upregulation of kynurenine and the activation of the aryl hydrocarbon receptor, which acts to metabolize melatonin and increase the N-acetylserotonin/melatonin ratio, with effluxed N-acetylserotonin acting as a brain-derived neurotrophic factor (BDNF) mimic via its activation of the BDNF receptor, TrkB, thereby increasing the survival and proliferation of tumors and cancer stem-like cells. This article highlights how many of the known regulators of cells in the tumor microenvironment can be downstream of the mitochondrial melatonergic pathway regulation. Future research and treatment implications are indicated.

## 1. Introduction

In recent decades, the conceptualization of cancers has focused on susceptibility mutations, with treatments determined largely by particular mutations evident in a given tumor. However, since the time of Warburg’s first publication on the role of metabolism and the shifting from oxidative phosphorylation (OXPHOS) to glycolysis [1], there has been a steady interest as to how metabolic processes drive cancer pathoetiology and pathophysiology. The Warburg effect is thought to arise from mutational and hypoxic challenges upregulating glycolysis for the provision of adenosine triphosphate (ATP), nucleotides, lipids, and amino acids for the growth of cancer cells even under aerobic conditions. More recent work shows many cancers to still predominantly use OXPHOS, such as epithelial ovarian cancer, although an upregulation of glycolysis is often evident [2]. A consequence of utilizing glycolysis as a primary energy source is that pyruvate is converted into lactate, which decreases the conversion of pyruvate to acetyl-CoA. Acetyl-CoA is necessary to optimize ATP production from the tricarboxylic acid (TCA) cycle, as well as OXPHOS. Acetyl-CoA is also a necessary co-substrate for the first enzyme in the melatonergic pathway, aralkylamine N-acetyltransferase (AANAT), indicating that the upregulation of glycolysis, and the relative suppression of acetyl-CoA, the TCA cycle and OXPHOS may be intimately associated with alterations in the regulation of the mitochondrial melatonergic pathway [3].

The tryptophan-melatonin pathway is shown in Figure 1 (gold shading), where the enzymatic regulators and necessary co-substrates of the tryptophan-melatonergic pathway are also shown (green shading). Melatonin is classically associated with its night-time release from the pineal gland, where it is associated with the regulation of the circadian rhythm, as well as being a powerful antioxidant, anti-inflammatory, antinociceptive, antineoplastic, and optimizer of mitochondrial function. The melatonergic pathway is initiated by the stabilization of AANAT by 14-3-3zeta in the presence of acetyl-CoA. Recent data indicates the melatonergic pathway to be evident in all body cells, mostly within mitochondria. Genomic regulation of the melatonergic pathway has been primarily investigated in the pineal gland, where the norepinephrine activation of the adrenergic Beta-1 receptor induces cAMP, which increases protein kinase (PK)A and the PKA phosphorylation of AANAT. The interactions of the melatonergic pathway with mitochondria and metabolism of tumors and tumor microenvironment cells are crucial aspects of the metabolic alterations long associated with tumors since the first publications by Warburg [3].

A shift to glycolysis is an evolutionary conserved process, a variation of which is an integral aspect of immune cell activation, which requires an upregulation of glycolysis during the course of immune cell activation, in order to energize and prepare immune cells entering a challenging environment. The incapacity to upregulate glycolysis, coupled to a low OXPHOS level, is what underpins the ‘exhaustion’ of natural killer (NK) cells and CD8+ t cells in the tumor microenvironment [3]. ‘Exhaustion’ refers to the failure of NK cells and CD8+ t cells to upregulate energy production (mostly glycolysis), which prevents these cells from producing a cytotoxic response, thereby preventing their capacity to kill cancer cells. Glycolysis upregulation is also necessary for macrophage activation, which invariably requires the induction of the inflammatory transcription factor, nuclear factor kappa-light-chain-enhancer of activated B cells (NF-kB). Interestingly, the process of macrophage activation by NF-kB also temporally synchronizes the deactivation of cells via NF-kB induction of the mitochondrial melatonergic pathway, and the subsequent effects of intracrine, autocrine and paracrine melatonin [4]. This sequential induction of the melatonergic pathway also seems an intimate aspect of the process of activation-deactivation in other, if not all, immune cells [5]. Such data in immune cells indicates that variations in relative glycolytic vs. OXPHOS metabolism, may be intimately linked to alterations in the regulation of the melatonergic pathway. This is strongly supported by the opposing effects of glycolysis and OXPHOS on the conversion of pyruvate to acetyl-CoA, given that acetyl-CoA is a necessary co-substrate for the initiation of the melatonergic pathway.

This article reviews the roles of metabolism in the different cells present in the tumor microenvironment, proposing that the wide array of diverse fluxes occurring among cells is strongly determined by core metabolic processes, of which the mitochondrial melatonergic pathway is an integral aspect. Cancer cells become the ‘dominant’ determinants of the tumor microenvironment by regulating the capacity of other cells present to induce the mitochondrial melatonergic pathway. NK cells and CD8+ t cells, rather than killing cancer cells, may conversely be providers of trophic support via the release of the melatonin precursor, N-acetylserotonin (NAS) [3]. The incorporation of the mitochondrial melatonergic pathway allows the better integration of the vast array of data collected on the tumor microenvironment and the intercellular interactions occurring, emphasizing the importance of core metabolic processes.

## 2. Tumor Microenvironment

Cancer cells do not exist independently in vivo but are part of a network of dynamically interacting cells within the tumor microenvironment.

### 2.1. Tumour Microenvironment Cells

A number of different cells can participate in the interactions of the tumor microenvironment, including cancer cells, cancer stem-like cells, macrophages, NK cells, CD8+ t cells, CD4+ t cells, dendritic cells, mesenchymal stem cells, γδ t cells, neutrophils, myeloid-derived suppressor cells (MDSCs), endothelial cells and platelets as well as quiescent or partially transformed resident cells in a given tissue [6]. These cells are in dynamic interaction, leading to variations in plasma membrane receptor and channel expressions, as well as releases and effects of thousands of proteins, including within exosomes and vesicles [6]. This has led to a plethora of studies over many decades detailing the alterations in different membrane expressed proteins and fluxes, as well as the intracellular factors regulating them, such as microRNAs (miRNA) and long non-coding RNAs. It is generally assumed that mutations have impacts on such plasma membrane expressions/fluxes/miRNAs, with the targeting of such mutations-driven plasma membrane proteins or fluxes being the primary treatment aim of many pharmaceuticals, including various immune-checkpoint inhibitors [7]. The complexity of such dynamic intercellular interactions has led to a large percentage of cancer funded research focusing on tumor cell lines, which emphasizes the importance of the individual tumor, whilst avoiding the complexity of tumor microenvironment interactions that occur in clinical cases.

More recent work has highlighted the importance of dynamic metabolism as an important aspect of the plasticity available to cells to deal with the ever-changing interactions within the tumor microenvironment [2]. Recently, it was proposed that tumors act to ‘domineer’ the other cells in the tumor microenvironment by shaping their mitochondrial function, especially via the regulation of the mitochondrial melatonergic pathway [3]. As this assumes that the mitochondrial melatonergic pathway is a core aspect of physiological function, it would be useful to review this pathway next.

### 2.2. Tryptophan-Melatonin Pathway

Most tryptophan is derived from dietary sources, although it is likely to that the presence of the Shikimate pathway in the human gut microbiome will also be a relevant tryptophan source [8]. Tryptophan is taken into cells via a number of different transporters, including the large amino acid transporter, (LAT)1 (SLC7A5), along with other essential amino acids. Tryptophan can be converted to serotonin by tryptophan hydroxylase (TPH)1 or TPH2. As TPH2 requires the 14-3-3eta isoform to be stabilized, factors regulating this 14-3-3 isoform will determine the capacity of cells to synthesize serotonin from tryptophan. Serotonin is then converted by 14-3-3zeta-stabilized AANAT, in the presence of acetyl-CoA, to N-acetylserotonin (NAS), which is then converted by acetylserotonin methyltransferase (ASMT) to melatonin. Factors regulating 14-3-3zeta and/or acetyl-CoA availability will then determine the levels of NAS and melatonin derived from serotonin. Not all NAS is converted to melatonin, leading to variations in the NAS/melatonin ratio, which is likely to be of some importance in the tumor microenvironment. Exogenous melatonin can be taken into mitochondria by the organic anion transporter (OAT)3 and peptide transporters (PEPT)1/2 [9]. See Figure 1.

A number of receptors with relevance in many cancers, can increase the NAS/melatonin ratio either by melatonin metabolism and/or via the ‘backward’ conversion of melatonin to NAS via O-demethylation, including the aryl hydrocarbon receptor (AhR), purinergic P2Y1 receptor, and the metabotropic glutamate receptor (mGluR)5 [10]. As 14-3-3 is suppressed by a number of miRNAs, including miR-7, miR-375, miR-451 and miR-709, these miRNAs will also dysregulate the tryptophan-melatonin pathway, in a manner dependent upon the 14-3-3 isoform suppressed [11,12]. These miRNAs are also associated with the regulation of different cancers [13,14,15], as are the P2Y1r [16] and mGluR5 receptors [17], whilst the AhR is a crucial aspect of cancers and the tumor microenvironment [18]. AhR activation suppresses NK cells and CD8+ t cells cytotoxicity and therefore their capacity to kill cancer cells [3,18]. The AhR is also activated by human epidermal growth factor receptor 2 (HER2) to drive mammosphere formation and cancer stem-like cell maintenance in breast cancers [19]. The AhR has a number of endogenous and exogenous ligands, including tryptophan-derived kynurenine.

The pro-inflammatory cytokines interleukin (IL)-1B, IL-6, IL-18 and tumor necrosis factor alpha (TNFa), but especially gamma-interferon (IFN-γ), induce indoleamine 2,3-dioxygenase (IDO) as well as stress/cortisol-associated tryptophan 2,3-dioxygenase (TDO), thereby converting tryptophan to kynurenine. Kynurenine and its metabolite, kynurenic acid, activate the AhR, indicating that the induction of these endogenous AhR ligands is intimately linked to suppressing tryptophan availability for the tryptophan-melatonin pathway. The AhR and melatonin also have reciprocal inhibitory interactions, including in NK cells where melatonin upregulates the cytotoxicity and cancer-killing capacity of NK cells [20], which is inhibited by the AhR, including via the AhR induction of the ‘immune checkpoint’ receptor, programmed cell death (PD)-1 [18]. The mutually antagonistic interactions of melatonin and the AhR are an important aspect of the intercellular interactions occurring in the tumor microenvironment.

Two key transcription factors associated with cancer promotion and the regulation of other cells in the tumor microenvironment are NF-kB [21] and yin yang (YY)1 [22]. Both NF-kB and YY1 upregulate the melatonergic pathway across different cell types [4,23], with the intracrine and autocrine effects of melatonin being built into the activation-deactivation process of NF-kB effects in reactive cells, such as macrophages [4]. The capacity of cancer associated AhR/CYP1B1, P2Y1r and mGluR5 to upregulate the NAS/melatonin ratio is important [24]. NAS is a brain-derived neurotrophic factor (BDNF) mimic via its activation of the BDNF receptor, TrkB [25]. TrkB activation by BDNF (or NAS) is a significant driver of cancer stem-like cell survival and proliferation [26]. TrkB expression is often indicative of poor prognosis across an array of different cancers [27]. Such data highlights the importance of variations in the melatonergic pathway in the tumor microenvironment, given that NAS, via TrkB activation, is associated with the proliferation and survival of cancer cells and cancer stem-like cells, whereas exogenous melatonin leads to the apoptosis of almost every cancer cell to which it has been applied [28]. This would strongly suggest that the regulation of the melatonergic pathway, especially the NAS/melatonin ratio, in the tumor microenvironment is of the utmost importance.

Exogenous, and presumably pineal, melatonin is taken up into tumor mitochondria via PEPT1/2, whilst the sulphation metabolites of melatonin are taken up via OAT3 [9]. Exogenous melatonin acts to induce the mitochondrial melatonergic pathway, often in association with an increase in OXPHOS and a reduction in Warburg-like glycolytic metabolism, as shown in ovarian cancer cells [29]. These authors showed melatonin to increase pyruvate and decrease lactate and lactate dehydrogenase, coupled to a positive correlation of ASMT with the pyruvate dehydrogenase complex (PDC) and ATP synthase genes, indicating the association of melatonin synthesis with OXPHOS [29]. These data are derived from an ovarian cancer cell line, SKOV3, and therefore are not necessarily reflective of the dynamic interactions that would be occurring in vivo. However, this data does highlight the significant impacts of exogenous and endogenous melatonin, and the dangers that this poses for cancer cell survival. A plethora of studies on an array of diverse cancers have shown similar endpoints, namely melatonin induced apoptosis in cancer cells, including via the suppression of PD-1 and the epithelial-mesenchymal transition (EMT) [30].

As all cells in the tumor microenvironment seem to have the capacity to produce melatonin, it would seem clear that for cancer cells to survive, they would have to suppress the capacity of neighboring cells to release melatonin and/or suppress melatonin effects. This seems to be primarily achieved by IDO/TDO induction and associated conversion of tryptophan to kynurenine, thereby activating the AhR to metabolize melatonin and increase the NAS/melatonin ratio. As shown in Figure 1, cancer cells may target a number of factors that act to regulate the mitochondrial melatonergic pathway in tumor microenvironment cells, including tryptophan uptake, 14-3-3 isoforms, TPH1/2, serotonin uptake/metabolism, acetyl-CoA, AANAT, ASMT, P2Y1r and mGluR5 receptor, as well as the AhR. Clearly, the diverse array of dynamic fluxes/exosomes/vesicles across cell types in the tumor microenvironment may be having significant impacts, both directly and indirectly, on the tryptophan-melatonin pathway.

The above begs the question as to why the melatonergic pathway is linked to core physiological processes. The melatonergic pathway is evident in all cells so far investigated, predominantly within mitochondria. This seems universal across the three kingdoms of life (plants, animals and fungi), due to the melatonergic pathway being evident in the ancient bacteria that crept into a unicellular organism to initiate the beginnings of multi-cellular life approximately 2 billion years ago [31]. The melatonergic pathway seems to have been maintained across the variety of life that has existed over the past 2 billion years, suggesting that it is an important aspect of life in its many manifestations. It is highly likely that the melatonergic pathway is evident in all cells of the tumor microenvironment. Consequently, the genetic or epigenetic suppression of the melatonergic pathway makes cells less capable of maintaining their homeostatic cellular function, making such cells less able to manage external influences, including the influence of cancer and cancer stem-like cells in the tumor microenvironment, allowing tumors to have a more dominant determination of intercellular interactions.

### 2.3. Mitochondrial Metabolism in Tumour Microenvironment

The factors underpinning alterations in mitochondrial function in the wide array of tumor microenvironment cells is an area of cutting-edge research.

#### 2.3.1. Natural Killer Cells

Optimally functioning NK cells have the capacity to kill cancer cells and virus-infected cells [32], which is dependent upon metabolism, namely the interactions of OXPHOS and glycolysis in NK cells [3]. Consequently, the regulation of NK cell activity, both directly and via other tumor microenvironment cells (see Figure 1), is crucial for tumor cell survival. Unlike cytotoxic t cells, NK cells do not require danger indicants/alarms, such as high-mobility group box (HMGB)1, ATP or heat shock protein (hsp)70 to detect and eliminate cancer cells. Rather, NK cell detection of major histocompatibility complex (MHC)-class 1 on cancer cell plasma membranes determines a cytotoxic response, involving the release of lytic enzymes, such as perforin and granzymes.

As indicated, a powerful aspect of NK cell control by the tumor is via IDO induction, thereby converting tryptophan to kynurenine, which activates the AhR [18]. As to whether the AhR-induced CYP1B1 leads to the ‘backward’ conversion of melatonin to NAS, with released NAS activating TrkB on cancer cells and cancer stem-like cells will be important to determine. If this occurs, it indicates that tumors turn their potential assassins into providers of trophic and metabolic support. The AhR-induced CYP1B1 also metabolizes melatonin, suppressing melatonin levels, thereby increasing the NAS/melatonin ratio.

NK cell cytotoxicity and IFN-γ production are intimately associated with optimized NK cell mitochondrial function [33], with acetyl-CoA potentially acting as an important interface between mitochondrial function and the capacity to induce glycolysis in the course of NK cell activation, at least partly via the acetylation of sphingosine, leading to N-acetylsphingosine, which acetylates and negates the suppressive effects of cyclooxygenase (COX)2/prostaglandin (PG)E2/EP2/4 and the deacetylated Raptor-induced mTORC1 mislocalization, thereby enhancing NK cell activation [3]. The maintenance of OXPHOS is therefore required to induce an effective, cytotoxic phenotype in NK cells. The suppression of NK cell OXPHOS and glycolysis is evident in cancer patients, which is at least partly driven by transforming growth factor (TGF)β [34] (See Figure 1). AhR activation not only increases COX2 and NAS, but also TGFβ, adenosine A2Ar and acetyl-CoA carboxylase, the latter decreasing acetyl-CoA availability. All of these AhR-driven changes suppress NK cell metabolism [3]. The consequences of these changes include alterations in levels of mitochondrial ROS and miRNA patterning, leading to an array of NK phenotype defining gene expressions, including the upregulation of PD-1. The details of such changes in NK cells will be important to determine, especially as to factors inducing miR-138, which suppresses PD-1 and PDK1 [35]. As melatonin also suppresses PDK1 [36,37], variations in the intracrine, autocrine and paracrine effects of melatonin will drive significant changes in the cytotoxicity of NK cells, and their capacity to kill cancer cells.

It is important to note that such changes in NK cell function do not seem restricted to NK cell responses to cancer, as the suppression of glycolysis, OXPHOS and mTORC1 [3] in circulating NK cells are evident in many other medical conditions, including inflammatory bowel disease [38], highlighting the ready plasticity of NK cell responses to the local cellular microenvironment. Melatonin optimizes NK cell cytotoxicity [39], including elevating the ageing-associated decrease in the plasma membrane expression of CD62L on NK cells, thereby enhancing NK cell (and CD8+ t cell) ability to enter sites of tumor growth and chronic inflammation [40]. As cancer is one of the many ageing-associated medical conditions [41], such impacts of melatonin will be relevant to how NK cell function modulates cancer pathoetiology and pathophysiology. The 10-fold decrease in pineal melatonin production from human adolescence to 80 years of age, will decrease the capacity of melatonin to optimize NK cell mitochondrial function and cytotoxicity efficacy over the circadian rhythm [42]. This may be important to cancer pathogenesis, given the increased risk of many cancers over the course of aging. As NK cell cytotoxicity is differentially regulated over the circadian rhythm, the attenuation of pineal melatonin production over the course of ageing, will contribute to suboptimal mitochondrial function and effective cytotoxicity in NK cells. This is likely to contribute to the course of ‘immune ageing’, which underpins many medical conditions, including cancer pathogenesis [3].

#### 2.3.2. Cytotoxic CD8+ t Cells

As with NK cells, CD8+ t cells can kill cancer cells and virus-infected cells. However, unlike NK cells, cytotoxic CD8+ t cells require priming by danger indicants/alarmins, such as HMGB1, ATP or hsp70 to detect and eliminate cancer cells [43]. Recent data has highlighted the crucial role of mitochondrial function, and translation, in determining CD8+ t cell serial cytotoxic responses to cancer cells and virus-infected cells following their initial activation by T cell receptors [44,45]. Like NK cells, the cytotoxicity of CD8+ t cells are significantly determined by the circadian rhythm [46].

CD8+ t cells are regulated similarly to NK cells in the tumor microenvironment, with CD8+ t cells also induced into a state of ‘exhaustion’. As with NK cells, CD8+ t cells are positively regulated by melatonin in cancers [47], as well as over the course of ageing, where melatonin offsets the ageing/inflammation-associated suppression of CD62L on CD8+ t cells [40]. Such data highlights how suppression of local and pineal melatonin will modulate CD8+ t cell cytotoxicity.

#### 2.3.3. Macrophages

M2-like macrophages are important regulators of the tumor microenvironment, significantly contributing to therapeutic resistance [48]. Macrophage depletion suppresses tumor growth and potentiates chemotherapy efficacy, as shown in lung cancers [49], indicating the significant role that M2-like macrophages have in the tumor microenvironment. As indicated above, macrophages are one of the few immune cells where the melatonergic pathway, via autocrine melatonin, is the major driver of the shift from an M1-like to an M2-like macrophage phenotype [4]. The macrophage melatonergic pathway can therefore have opposing effects on M1-like and M2-like macrophages phenotypes in the tumor microenvironment, being apoptotic and pro-growth, respectively [49]. However, should macrophages release melatonin into the tumor microenvironment, this could prove problematic to cancer cells and cancer stem-like cells, given the toxic effects of melatonin in cancers.

As noted, COX2/PGE2/EP2/4 activation are significant contributors to NK cell ‘exhaustion’ [3]. COX2-derived PGE2 is also a significant determinant of CD8+ t cell ‘exhaustion’ in the tumor microenvironment, with effects that seem mediated via macrophages [50]. These authors propose that it is the release of macrophage TGFβ that drives CD8+ t cell [50], and NK cell [34], exhaustion. Interestingly, PGE2 in macrophages drives an M2-like phenotype via the upregulation and induction of the BDNF/TrkB pathway [51]. This would indicate that the induction and release of NAS, via autocrine effects at TrkB, may also contribute to the M2-like macrophage phenotype in the tumor microenvironment, with consequent increases in TGFβ release that contributes to ‘exhaustion’ in NK cells and CD8+ t cells. Concurrently, macrophage released NAS would have paracrine effects, including activating TrkB on cancer stem-like cells to enhance their proliferation and survival.

The interactions of NK cells, CD8+ t cells and macrophages in the tumor microenvironment provides a simple model of how intercellular communication among these cells may be importantly driven by variations in their mitochondrial melatonergic pathway, as shown in Figure 2.

However, the dynamic interactions of tumor microenvironment cells will considerably complicate their intercellular communications. For macrophages to produce and release NAS would require either a suppression of its conversion to melatonin and/or the conversion of melatonin to NAS by the AhR, P2Y1r or mGluR5 [3]. As noted, the AhR is an important target for tumor-derived kynurenine in the suppression of NK cell and CD8+ t cell activation [18]. However, the consequences of AhR effects are highly variable, partly dependent upon its many specific ligands and the particular cell being investigated. Gut microbiome-derived indole-3-propionate partially activates the AhR on macrophages to induce an M2-like phenotype [52], similar to the macrophage phenotype in the tumor microenvironment. AhR activation by air pollution particular matter elevates heparin-binding EGF-like growth factor (HBEGF) in macrophages, the release of which induces epithelial-to-mesenchymal transition (EMT) in cancer cells, thereby increasing metastasis [53]. However, in fibrotic pulmonary epithelial cells, AhR activation can trigger a pro-inflammatory M1-like macrophage phenotype [54], highlighting the complex, and often contrasting, consequences of AhR activation. Such complexity seems to arise from variations in the activation of the macrophage mitochondrial melatonergic pathway. As indicated, COX2-derived PGE2 induces BDNF and TrkB activation, leading to an M2-like macrophage phenotype, as would NAS [51]. This could indicate that factors such as COX2/PGE2/BDNF-NAS/TrkB activation, in driving alterations in core mitochondrial processes, will be contributing to the seeming complexity of AhR activation in macrophages, and in other cell types. Clearly, the interactions of melatonergic pathway regulation with AhR effects requires investigation in macrophages of the tumor microenvironment.

Macrophage activation typically involves NF-kB induction to induce a pro-inflammatory phenotype patterning of induced genes. NF-kB also upregulates melatonin production, the autocrine effects of melatonin providing time-limited inflammatory responses [4]. As to how the inability of NF-kB to induce NAS and/or melatonin interacts with the activation and deactivation of macrophages will be important to determine. Other factors acting to suppress macrophage activation and deactivation, such as specialized pro-resolving mediators, will also require investigation in this context. Clearly, the interaction of the AhR with NF-kB and the melatonergic pathway in the regulation of macrophage, and other immune cell, require investigation.

Other regulators of the NAS/melatonin ratio may also be relevant in the modulation of macrophage plasticity in the tumor microenvironment. Extracellular ATP levels are higher in the tumor microenvironment [55], although can be quickly converted to adenosine. M1-like and M2-like macrophages can be differentiated by their expression of purinergic receptors, with P2Y13r and P2Y14r overexpressed in the M1-like phenotype, whilst the P2Y1r and P2Y6r are exclusively upregulated in the M2-like phenotype [56]. This would indicate a role for P2Y1r activation inducing an increase in the NAS/melatonin ratio in the maintenance of macrophage phenotypes in the tumor microenvironment, concurrent to a maintained provision of macrophage-derived NAS for TrkB activation on cancer stem-like cells. Glutamate can also be abundantly released from tumor cells, contributing not only to the suppression of neutrophil killing capacity in the tumor microenvironment [57], but also to mGluR5 activation. Activation of mGluR5 also elevates the NAS/melatonin ratio, with macrophage mGluR5 activation initiating a metabolic rearrangement in macrophages that is proposed to contribute to an immunosuppressive phenotype [58]. The heightened levels of tumor-derived glutamate and ATP, like kynurenine, may be intimately associated with the regulation of the NAS/melatonin ratio across different cells of the tumor microenvironment.

Overall, the ready availability of ligands for the AhR, P2Y1r, and mGluR5 in the tumor microenvironment, and the consequences of their activation in macrophages suggest significant impacts on metabolism that include the upregulation of the NAS/melatonin ratio, contributing to an M2-like phenotype, whilst possibly enhancing the availability of NAS to promote cancer stem-like cell survival and proliferation. The production and release of NAS will be important to determine in macrophages, including as to the relevance of macrophage NAS in the maintenance, proliferation, migration and re-emergence of cancers.

However, as the removal of macrophages from the tumor microenvironment limits, but does not stop, progression [49], it is clear that other tumor microenvironment cells have significant impacts.

### 2.4. Neutrophils

Neutrophils are often defined as the ‘fast-responding’ immune cells to challenge or invasion. Recent work has indicated a role for neutrophils in the tumor microenvironment [59]. Neutrophils are directly, and indirectly, regulated by AhR activation [60]. Neutrophils also express mGluR5, which modulates their activity [61], whilst neutrophils are also a significant source of extracellular glutamate [62]. P2Y1 receptor activation on platelets [62] and macrophages [63] allows these cells to chemoattract neutrophils in the course of inflammation-associated medical conditions. Such data would indicate that alterations in melatonergic pathway regulation may be important aspects of neutrophil chemoattraction and activation.

Neutrophils have relatively few mitochondria, relying more on glycolysis for energy production [64]. This has led to a paucity of research on the role of mitochondria in neutrophils. However, recently this has changed, with a growing body of evidence indicating the significant role that mitochondria play in neutrophil development, chemotaxis, effector function/phenotype and apoptosis, as well as emerging treatment implications [65]. Autocrine mitochondrial ATP at the P2Y2r is necessary for neutrophil chemotaxis, and mitochondrial ROS is necessary for effector functions [64].

Exogenous melatonin has significant impacts on neutrophils, including as to how neutrophils regulate breast cancer cell survival and apoptosis [66]. The beneficial effects of exogenous melatonin in optimizing neutrophil effector function and ROS-driven killing capacity are attributed to melatonin upregulation of neutrophil glutathione (GSH), highlighting the importance of antioxidant regulation in neutrophil function [67]. Neutrophil GSH upregulation is dependent upon heightened expression of the cystine-glutamate antiporter [68], indicating that neutrophil activation will be intimately linked to the paracrine and autocrine effects of glutamate. In the course of sepsis, the beneficial effects of melatonin are mediated via the promotion of neutrophil extracellular trap (NET), coupled to an inhibition of a neutrophil phagocytic (N2-like) phenotype [69]. The N2-like neutrophil phenotype participates in inflammation resolution and phagocytic activity, which includes the upregulation of BDNF effects at the TrkB receptor [70]. Both NF-kB [71] and YY1 [72] regulate neutrophil activation and chemotaxis. As both NF-kB and YY1 concurrently upregulate the melatonergic pathway, this would be indicating a role for melatonergic pathway availability, especially the NAS/melatonin ratio, in determining neutrophil phenotype and effects within the tumor microenvironment. Although there is no data pertaining to the presence or regulation of the melatonergic pathway in neutrophils, the data highlighted above would indicate that variations in the autocrine and paracrine effects of NAS *vs* melatonin would have distinct impacts on neutrophil phenotype and interactions with other cell types. Factors regulating the tryptophan-melatonin pathway, including 14-3-3 proteins and serotonin (as shown in Figure 1), will be important to determine in neutrophils. This will clarify the relevance of the neutrophil tryptophan-melatonin pathway, including how this pathway is shaped by intercellular interactions within the tumor microenvironment.

### 2.5. Platelets

Recent data shows platelets to have a significant role in the tumor microenvironment, including via their interactions with circulating neutrophils [73]. The tumor microenvironment induces platelet aggregation and activation. Platelets are therefore a significant treatment target in cancer management [74]. Platelets are involved in multiple processes in the tumor microenvironment, including angiogenesis [75], microvesicle release [76], and PD-L1 induction in cancer cells via NF-κB and/or TGFβ [77], with platelet-derived subtypes proposed to regulate metastasis [78], including via lncRNAs [79] and miRNAs [80]. P2Y1 receptor activation on platelets allows platelets to chemoattract neutrophils in the pathophysiology of a wide array of inflammation-associated medical conditions [62], including in LPS induced pulmonary disorders [81]. The AhR is a significant regulator of platelet development [82] and primed activation [83], indicating a role for cancer cell kynurenine release in platelet activation. The impact of these melatonergic pathway regulating factors, P2Y1r and AhR, on platelet function would indicate the relevance of the platelet mitochondrial melatonergic pathway.

Platelets are a significant source of serotonin in the tumor microenvironment. Preclinical data show platelet-derived serotonin to contribute to CD8+ t cell suppression and raised levels of PD-1, including in human cancer cells [84]. Serotonin concentrations in the metastases of cancer patients inversely correlates with CD8+ t cell levels. Serotonin depletion in preclinical models reverses these effects as well as decreasing PD-1 [84]. Such data indicates that the controversial role of platelets in the tumor microenvironment may be via its serotonin supply as a ready substrate for NAS synthesis, thereby suppressing apoptosis, and increasing proliferation as well as driving TGFβ in macrophages, and perhaps other cells of the tumor microenvironment. The importance of platelets in the tumor microenvironment would therefore be intimately linked to the NAS/melatonin ratio, with consequences for intercellular fluxes and coordinated immune responses, including via NAS/TrkB/TGFβ in macrophages driving a tolerogenic dendritic cell phenotype that suppresses CD8+ t cells and NK cells.

Mitochondria-derived ROS is a significant determinant of platelet activation [85], with effects that include the induction of ROS-dependent miRNAs, thereby impacting on gene patterning. Platelet mitochondria can also be released in extracellular vesicles to modulate neighboring cell metabolic function [86]. Platelet deactivation involves sirtuin-1 and sirtuin-3 activation, with associated ROS suppression and optimized mitochondrial function [87]. Pyruvate, TCA cycle regulation and OXPHOS capacity are significant determinants of platelet activation as well as the contents of excreted granules [88]. As with immune cells, platelet activation involves the upregulation of aerobic glycolysis, coupled to maintained/upregulated OXPHOS [89], suggestive of a synchronized alteration in the regulation of the platelet mitochondrial melatonergic pathway.

Exogenous melatonin suppresses platelet activation [10], as well as affording protection against platelet apoptosis [90]. High levels of serotonin are taken up and released by platelets, indicating that platelets may be a significant source of serotonin as a necessary precursor for the initiation of the melatonergic pathway. However, unlike serotonin, melatonin does not seem to be stored in platelets [91]. As to whether the platelet melatonergic pathway can be induced and/or upregulated to utilize stored serotonin or serotonin taken up into platelets will be interesting to determine. Importantly, platelets store very high BDNF concentrations, with BDNF acting to prime platelet activation, whilst very high, but feasible, BDNF concentrations drive platelet aggregation [92]. Platelets express a truncated form of TrkB (TrkB-T), both on their plasma membrane and within an intracellular compartment, with TrkB-T, lacking a tyrosine kinase domain [92]. TrkB-T activation enhances platelet activation and aggregation, including the release of pro-inflammatory and pro-angiogenesis factors [92]. This is suggestive of a significant role for circulating BDNF in platelet activation. However, this data would also indicate that platelet NAS, as well as NAS derived from other cells in the tumor microenvironment, would directly act on TrkB-T, to upregulate pro-angiogenic and inflammatory cytokines. NAS may also upregulate BDNF, as shown in the murine hippocampus [93]. Platelet TrkB-T activation is proposed to mediate its effects via Rac1/PKC/Ca^2+^ as well as STAT3 [92].

However, the array of platelet effects may ultimately be driven by changes in the platelet mitochondrial melatonergic pathway in association with heightened AhR activation by tumor-derived kynurenine. As to whether kynurenine/AhR/CYP1B1 drive an increase in the NAS/melatonin ratio in tumor-associated platelets will be interesting to determine. This would also suggest that under conditions of suppressed tryptophan availability for serotonin synthesis, including from IDO/TDO conversion of tryptophan to kynurenine, platelets provide a ready supply of serotonin. Whether platelet-provided serotonin differentially regulates the mitochondrial melatonergic pathway of tumor microenvironment cells, by having its consequences limited to cells able to uptake serotonin and convert it to NAS will be interesting to determine. Should such differential serotonin uptake and conversion to NAS occur, this would have significant impacts on the intercellular interactions within the tumor microenvironment.

This could indicate a significant role for variations in the NAS/melatonin ratio in the regulation of platelet function in the tumor microenvironment, driven by the contrasting effects of NAS and melatonin. Platelets may therefore be significant regulators of NAS (and BDNF) production in the tumor microenvironment, with consequences for angiogenesis and wider inflammatory-driven processes. Humans express two truncated TrkB receptors, TrkB-T1 and TrkB-shc [92]. Data on TrkB-T1 indicate that it is upregulated by mitochondrial ROS inducing ROS-dependent miRNAs, such as miR-4813, miR-34a and down regulated by miR-185-3p [94], which may also be important in the pathoetiology of amyotrophic lateral sclerosis [95]. As to whether the priming effect of tumor-derived kynurenine on AhR activation in platelets contributes to variations in the platelet NAS/melatonin ratio, in turn driving an increase in TrkB-T levels, activation and ligands, will be important to determine. Clearly, NAS and its cellular sources in the tumor microenvironment require investigation, especially as the proposed role of NAS in proliferative conditions has long been indicated [96], including in cancers [97].

### 2.6. Other Tumour Microenvironment Cells

A number of other cell types are commonly evident in the tumor microenvironment, including myeloid-derived suppressor cells (MDSC). MDSC, as well as M2-like macrophages, γδ t cells, cancer-associated fibroblasts and Regulatory t cells (Treg) may all release TGFβ, thereby contributing to the exhaustion and immune suppression evident in the tumor microenvironment [3,98]. As noted previously, the PGE2 induction of the BDNF/NAS-TrkB pathway in M2-like macrophages is associated with TGFβ upregulation and efflux, classically associated with the subsequent ‘exhaustion’ in NK cells and CD8+ t cells [51]. The role of the mitochondrial melatonergic pathway, specifically NAS, in MDSCs, γδ t cells and Treg cells in the induction and release of TGFβ will be important to determine.

#### 2.6.1. Myeloid-Derived Suppressor Cells (MDSCs)

MDSCs are a heterogeneous collection of immature myeloid cells that are strongly associated with immune suppression. MDSCs release adenosine in the tumor microenvironment, thereby activating the adenosine A2Ar in cytolytic cells to suppress their function. Importantly, MDSCs phenotype and function are powerfully determined by metabolic processes [99]. As AhR activation lead to dramatic mobilization of MDSCs in the peritoneal cavity coupled to enhanced immune suppression, increased glycolysis, and mitochondrial respiration, coupled to alterations in miRNA patterning [100], it requires investigation as to the role of AhR-induced CYP1B1 and alterations in the regulation of the melatonergic pathway in MDSCs, including as to MDSCs melatonergic pathway regulation by tumor-released kynurenine and gut microbiome-derived AhR ligands, such as indole-3-propionate [101].

#### 2.6.2. Regulatory t Cells (Treg)

The AhR is intimately associated with the induction of Treg and PD-1 upregulation [102], indicating tumor-derived kynurenine and gut tryptophan metabolites in the co-ordination of Treg and CD8+ t cell exhaustion. There is no data on any relevant roles for mGluR5 or P2Y1r in Treg or MDSCs. However, Treg do express tryptophan hydroxylase (TPH)1, and therefore have a serotonin synthesis capacity following tryptophan uptake as do, to a lesser degree, Th1 and Th17 cells [103]. As to whether such synthesized serotonin is utilized as a precursor for the mitochondrial melatonergic pathway in Treg requires investigation. There is some evidence in hepatocellular carcinoma of exosomal transfer of 14-3-3 to t cells, leading to a decrease in effector t cells and increase in Treg [104]. As to whether this increases the stabilization of AANAT, leading to melatonergic pathway induction and alterations in the NAS/melatonin ratio will be interesting to determine, including as to the relevance of the NAS/melatonin ratio to TGFβ release in Treg. Interestingly, melatonin decreases the differentiation of naïve t cells into Treg, and therefore suppresses levels of TGFβ efflux and TGFβ-induced exhaustion of cytolytic cells [105]. This highlights the importance of the local regulation of the melatonergic pathway across cell types, including Treg, in shaping patterned immune responses in the tumor microenvironment.

#### 2.6.3. γδ t Cells

γδ t cells have been relatively little explored in comparison to other immune cells. γδ t cells have a number of different phenotypes, including IFN-γ or IL-17 producing (γδ17 t cells) phenotypes, with γδ17 t cells playing a pro-tumoral role [106]. γδ17 t cells are associated with an increase in ROS and glycolytic metabolism [106]. As to whether this is accompanied by alterations in the regulation of the mitochondrial melatonergic pathway is unknown. γδ t cells can also have a distinctive ‘exhausted’ phenotype in the tumor microenvironment in association with increases in PD-1 expression [107]. Gut microbiome-derived histone deacetylase (HDAC) inhibitors suppress IL-17 production from γδ17 t cells [108], indicating a significant role of the gut microbiome in the regulation of γδ17 t cell effects and interactions in the tumor microenvironment.

#### 2.6.4. Dendritic Cells

Dendritic cells (DCs), by activating antigen-specific T cells, can enhance the antitumor effects of t cells. DCs are also important in tumor progression from premalignant state [109]. DCs are an important interface between the innate and adaptive immune systems. DC function is significantly determined by variations in mitochondrial function, with obesity-associated deficits in DC mitochondrial function suppressing the antigen-presenting capacity of these cells in association with heightened ROS production [110]. As these changes in DC in obese patients can be rectified by antioxidants [110], the regulation of the DC mitochondrial melatonergic pathway will be important to determine, including in regard to the heightened risk and accelerated pathophysiology of cancer in obesity. As the conversion of pyruvate to acetyl-CoA and OXPHOS are significant determinants of DC activation and antigen-presenting capacity [111], it will be important to determine the relevance of the mitochondrial melatonergic pathway in these cells, not only in cancers but in many other medical conditions. AhR activation in DCs leads to tolerogenic DCs, which can increase Tregs [112]. TGFβ also induces a tolerogenic DC phenotype, and therefore suppressed priming for activation of CD8+ t cells [113]. (See Figure 1). Melatonin powerfully regulates DC function, including suppressing the ROS and enhancing endogenous antioxidants [114], thereby preventing the suppressive effects of ROS on DC antigen-presenting capacity. As TrkB seems to have limited, if any, expression in human DCs [115], any increase in the extracellular NAS/melatonin ratio is likely to significantly impact on DC function via NAS/TrkB and how these cells drive the interactions of innate and adaptive immune systems. Cancer cell-derived kynurenine activation of the AhR, driving a decrease in melatonin will increase tolerogenic Tregs, whilst the loss of melatonin optimization of DC function contributes to suppressed activation of CD8+ t cells in the tumor microenvironment.

#### 2.6.5. Mesenchymal Stem Cells

Mesenchymal stem cells (MSCs) are stromal cells that can self-renew as well as having the capacity to differentiate into multiple lineages and efflux large quantities of exosomes [3]. MSCs are present in almost all tissues, where they participate in tissue regeneration and homeostasis, involving dynamic interactions with other cells that are dependent on cell phenotype [116]. Given such physiological effects, MSC are important tumor microenvironment regulators, including via their capacity to increase tumor PD-L1 expression [117]. The capacity of MSC to release mitochondria, including within vesicles, as well as regulating the mitochondrial function of other cells, makes MSC a significant treatment target. Mitochondrial transfer into rodent tumors significantly modulates tumor progression and patterned immune responses [118], highlighting the importance of core metabolic processes in the intercellular interactions of the tumor microenvironment.

Melatonin is a significant regulator of MSC OXPHOS and sirtuins, as well as the content of exosomes and vesicles, indicating the important role of alterations in the melatonergic pathway in the tumor microenvironment in determining MSC intercellular effects [3]. BDNF at TrkB regulates MSC development, whilst the capacity of MSC to secrete BDNF will have significant impacts in the tumor microenvironment. AhR activation is a significant regulator of the many functions of MSC, including suppressing the proliferation of bone marrow-derived MSCs, whilst enhancing mitochondrial function and mitochondrial transcription factor A (TFAM) [119], indicating the tumor-derived kynurenine, via AhR activation, will better optimize the mitochondrial function of MSC and attenuate MSC proliferation. Given the AhR metabolizes melatonin and will increase the NAS/melatonin ratio, it requires investigation as to whether the pre-existent state of the MSC melatonergic pathway determines the consequences of AhR activation.

### 2.7. Melatonergic Pathway and Immune Cells

As indicated above, exogenous (and presumably pineal-derived circadian) melatonin has significant impacts on tumor microenvironment immune and tumor cells, often with some contrasting effects compared to TrkB activation by BDNF. However, to date the role of the melatonergic pathway in tumor-relevant immune cells has only been investigated in macrophages [4] and, in the case of CNS cancers, in microglia [5]. This urgently requires rectification, given the apparent importance of the melatonergic pathway on immune cell function and consequences of alterations in the melatonin/NAS ratio. The lack of investigation regarding the mitochondrial melatonergic pathway in immune cells seems to be a historical legacy in the defining of immune cell phenotypes by their expression of plasma membrane receptors.

The roles of the melatonergic pathway and AhR in the tumor microenvironment are given support by recent data derived from 33 different types of solid tumors, showing the significance of increased CYP1B1 in the regulation of immune responses in the tumor microenvironment and their association with patient clinical characteristics [120]. These authors showed CYP1B1 to be correlated with tumor grade, clinical stage, immune cell subtype infiltration, tumor mutation burden, and DNA sequence microsatellite instability, as well as therapeutic resistance. Such changes are proposed to indicate CYP1B1 as a significant treatment target [120,121]. As noted above, CYP1B1 is downstream of AhR activation, being associated with the upregulation of the NAS/melatonin, partly mediated via the 6-hydroxylation of melatonin, although melatonin can also undergo O-demethylation in association with an enhanced NAS/melatonin ratio [122]. Most of the effects of CYP1B1 across the 33 types of solid tumors investigated in this study, were reversed by the application of exogenous melatonin, including angiogenesis marker CD31 expression, ki67 (proliferation marker), matrix metallopeptidase (MMP)9, as a metastasis marker, and LY6G (neutrophil infiltration marker) [120]. Such data is supportive of the wide-ranging consequences of tumor microenvironment interactions being powerfully determined by variations in the mitochondrial melatonergic pathway across cell types.

### 2.8. MicroRNAs and Tumour Microenvironment

Alterations in numerous miRNAs have been associated with tumor pathophysiology, as would be expected given their role in determining patterned gene expression. A number of miRNAs are strongly associated with the regulation of the melatonergic pathway across different cell types, including miR-7, miR-375, miR-451 and miR-709 [3,123,124,125,126]. All of these miRNAs can regulate 14-3-3 isoforms, and therefore can impact on the melatonergic pathway via the suppression of 14-3-3eta and 14-3-3zeta, which are the 14-3-3 isoforms that stabilize TPH2 and AANAT, respectively. As with the exosomal transfer of 14-3-3 from hepatocellular carcinoma cells to t cells [104], variations in these miRNAs in the different cells of the tumor microenvironment, via 14-3-3 isoform regulation, can regulate the mitochondrial melatonergic pathway. Consequently, exosomal miRNAs, as well as 14-3-3 isoforms, are another means by which cancer cells can regulate mitochondrial melatonergic pathway in cells of the tumor microenvironment, thereby not only impacting on NAS and melatonin production, but also on mitochondrial ROS, ROS-driven miRNAs and patterned gene expression. All of these miRNAs have been linked to a plethora of gene variations that have been proposed to drive their effects in the tumor microenvironment. However, as indicated throughout, the impact of these miRNAs in different cells of the tumor microenvironment may be importantly determined by their influence on core mitochondrial function, including the mitochondrial melatonergic pathway.

### 2.9. O-Linked-N-Acetylglucosaminylation (O-GlcNAcylation) and Tumour Microenvironment

O-GlcNAcylation is a form of glycosylation, where the monosaccharide, OGlcNAc, is added to a serine or threonine residue of nuclear and/or cytoplasmic protein by O-GlcNAc transferase (OGT), which can be readily reversed by O-GlcNAcase (OGA). O-GlcNAcylation can significantly regulate cells and important transcription factors in the tumor microenvironment, including the AhR, YY1, the NLRP3 inflammasome, MDSCs, NK cells, and CD8+ t cells, as well as sirtuins and core metabolic processes [3]. The association of O-GlcNAcylation may therefore be intimately linked to mitochondrial melatonergic pathway regulation.

## 3. Tumor Mitochondria as Conductors of the Tumor Microenvironment Mitochondrial Orchestra

Although alterations in metabolism have been appreciated as significant regulators of tumor cell function and patterned gene expressions since the first publication by Warburg a century ago, there is a growing appreciation that mitochondria are the major determinant of coordinated cellular function, including in immune cells. The tumor microenvironment can be conceived as a ‘domineering’ influence of tumor cells and cancer stem-like cells in shaping the interactions of other cells in this environment by acting to regulate mitochondrial function in these cells. By suppressing the ability of cells to self-determine previously established homeostatic regulation, tumors conduct this orchestra of cells to their preferred tune, namely the survival and proliferation of tumor cells and cancer stem-like cells. This is primarily achieved via mitochondrial melatonergic pathway regulation.

As melatonin induces apoptosis in all cancer cells to which it is applied, it is clearly in the vested interest of the tumor to suppress the production and release of melatonin in the tumor microenvironment. This can be achieved via tumor induction of IDO/TDO and the conversion of tryptophan to kynurenine, which when released by the tumor activates the AhR to metabolize melatonin, whilst the O-demethylation of melatonin to NAS, allows this BDNF mimic to activate TrkB, thereby increasing the proliferation and survival of cancer cells and cancer stem-like cells. Such effects make the mitochondrial melatonergic pathway a crucial target for the tumor to shape the microenvironment mitochondrial orchestra to its own requirements. Although requiring investigation in most of the cells of the tumor microenvironment, all cells investigated to date show the presence of a melatonergic pathway, mostly evident in mitochondria, but also to a much smaller degree in the cytoplasm.

Optimized mitochondrial function is generally associated with OXPHOS, which is importantly determined by the conversion of pyruvate to acetyl-CoA, thereby increasing ATP production by the TCA cycle as well as being a necessary co-substrate for AANAT and the initiation of the melatonergic pathway. The tumor can regulate a number of factors to suppress the melatonergic pathway in other cells, including suppressing tryptophan availability, and tryptophan uptake, as well as the conversion of tryptophan to serotonin, which requires TPH, with TPH2 requiring 14-3-3e for its stabilization. TPH, and 14-3-3e may therefore be other direct and/or indirect targets of the tumor in other cells of the tumor microenvironment. The conversion of serotonin to NAS by AANAT requires the presence of acetyl-CoA and 14-3-3z, making the regulation of acetyl-CoA and 14-3-3z other targets to limit the melatonergic pathway in tumor microenvironment cells. As the tumor benefits from the BDNF-mimicking effects of NAS, factors that increase the NAS/melatonin ratio, including the AhR, P2Y1r and mGluR5 become other direct and indirect targets of the tumor in its quest for survival.

As highlighted above, although requiring investigation in most tumor microenvironment cells, it is clear that alterations occur in these cells in factors known to regulate the mitochondrial melatonergic pathway. From the tumor’s perspective, the many undesirable effects of melatonin can occur in most of the cells of the tumor microenvironment, such as melatonin increasing dendritic cell function and antigen-presenting capacity, and thereby enhancing CD8+ t cell cytotoxicity against the tumor. Given that the intracrine and autocrine effects of melatonin regulate endogenous antioxidants, OXPHOS and gene patterning, the suppression of melatonin production limits the capacity of these cells to homeostatically self-regulate. Presumably, this is a long-established evolutionary effect that allows particular cells to determine the nature of the homeostatic status quo to be established, and may well date back to 2 million years ago when ancient bacteria in a single cell organism interacted with other bacteria in a single cell organism in the etiology of multi-cellular life [31], which may have contributed to the shaping of different cell types.

An important aspect of attenuated melatonin production is the increase in mitochondrial ROS, which drives ROS-regulated miRNAs, thereby impacting patterned gene expression. Consequently, the suppressed capacity to upregulate the melatonergic pathway in a given cell in the tumor microenvironment leads to a dramatic change in the genes induced and thereby in the function of the given cell. Some of the detailed consequences have been indicated above, and previously [3]. Given that the first ancient bacteria to creep into a cell-like structure 2 billion years ago seems to have been a melatonin-producing ancient bacteria, and that the melatonergic pathway seems to be evident in all 3 kingdoms of multicellular life on earth (animals, plants and fungi), the melatonergic pathway is clearly a crucial aspect of the capacity of multi-cellular organisms to maintain their survival over the course of diverse evolutionary pressures. It is this core aspect of evolved physiological function that the tumor seeks to control. The plethora of intra- and inter-cellular fluxes that have been investigated in the tumor microenvironment may be seen as a consequence of dysregulated mitochondrial melatonergic pathway activity in the course of the tumor establishing a new homeostatic status quo favoring its survival and proliferation.

Although some details are highlighted above to strengthen the case for the importance of the melatonergic pathway across cell types in the tumor microenvironment, it is clear that the melatonergic pathway needs extensive investigation across different cell types. For example, does the ten-fold decrease in pineal gland night-time melatonin production between the ages of 18 years and 80 years of age occur in other cell types? If so, does this then change the capacity of a given cell to maintain an established homeostatic interaction with other interacting cells over the course of ageing? There seems a never-ending pursuit of looking at fluxes in cancers and the avoidance of core processes driving such fluxes. This may be most typified in the investigation of immune cells and their classification on the basis of plasma membrane receptor/channel/protein expression. This ‘judging the book by its cover’ has avoided the challenge of looking at fundamental core processes for many decades, which has been changing in recent years. Hopefully, the fact that melatonin kills all cancers to which it is applied, would indicate that the melatonergic pathway should be a priority for investigation in the regulation of metabolism in all cells of the tumor microenvironment.

It should be noted that there has been some limited investigation of the melatonergic pathway in cancer-like cell lines indicating that the accumulation of NAS in mitochondria induces apoptosis in these cells [127]. Such data indicates the presence of the melatonergic pathway in tumor-like cells and the significant impact that alterations in this pathway can have, via mitochondrial CYP1B1 driving an increase in the mitochondrial NAS/melatonin ratio in tumor cells [127]. As to whether this occurs in vivo in the complexity of the tumor microenvironment will be important to determine, including as to the relevance of exogenous melatonin’s conversion to NAS in the mitochondria of tumors. It would be interesting if the NAS intracellular accumulation is toxic, whilst its extracellular effect at TrkB is trophic in tumors. It would also suggest that the regulation of melatonin uptake into mitochondria, via PEPT1/2 and OAT3, may be very important to cancer cell survival.

Overall, this could suggest, in the footsteps of tumors, that targeting the melatonergic pathway in tumors will give clinicians an important control over the tumors influence on other cells in the tumor microenvironment. However, it may be more practical to target the melatonergic pathway in other cells in the tumor microenvironment, such as natural killer cells and/or dendritic cells, likely involving antagonism of the AhR and probably other targets identified by future research in this crucial area of core physiological function. The above provides a frame of reference that allows the incorporation, and reframing of effects, of a number of genetic, epigenetic and environmental factors relevant to tumor pathoetiology and pathophysiology.

## 4. Future Research

Of fundamental importance, is NAS effluxed from any tumor microenvironment cell to activate TrkB on cancer cells and cancer stem-like cells?Is the melatonergic pathway evident in all cells of the tumor microenvironment?Glyphosate-based herbicides (GBH) have recently been associated with an increased risk of cancer, especially subtypes of non-Hodgkin’s lymphoma. Recent work indicates that GBH may significantly contribute to the pathoetiology and pathophysiology of amyotrophic lateral sclerosis [95], with effects relevant to alterations in the mitochondrial melatonergic pathway, both directly and via gut dysbiosis. Is GBH exposure relevant more widely in cancer pathophysiology?Does melatonin inhibit PD-1 [30] via the suppression of miRNAs that induce PD-1 and/or the induction of miRNAs that suppress PD-1, such as miR-138? As miR-138 also suppresses PDK1, and therefore increases PDC, in some cells [35] this would suggest that the impacts of melatonin in mitochondria will determine ROS-driven miRNA patterning, and therefore have an impact on crucial patterned gene inductions.Does the COX2/PGE2/BDNF-NAS/TrkB activation pathway, in driving alterations in core mitochondrial processes, underpin the complex variations in the effects of AhR activation in macrophages, and in other cell types? Is the complexity of AhR effects, and the mixed results arising, intimately linked to unmeasured variations in the melatonergic pathway?Are the AhR interactions with NF-kB and the melatonergic pathway relevant to variations in macrophage phenotype induction in the tumor microenvironment?Is the melatonergic pathway evident in neutrophils, including within the cytoplasm and/or mitochondria? Would the autocrine and paracrine effects of NAS vs. melatonin significantly regulate the neutrophil phenotype and neutrophil interactions with other cells in the tumor microenvironment?How are melatonergic pathway regulating factors, such as 14-3-3 proteins, serotonin, LAT1 and acetyl-CoA in neutrophils modulated by intercellular processes in the tumor microenvironment?Can the platelet melatonergic pathway be induced and/or upregulated to utilize stored serotonin or serotonin taken up into platelets?What is the role of the mitochondrial melatonergic pathway and CYP1B1, following AhR activation, in the dramatic induction of MDSCs [100]?Do all cancer cells transfer 14-3-3zeta in exosomes to t cells in order to upregulate AANAT, which, when coupled to kynurenine activation of the AhR, increases the NAS/melatonin ratio, as indicated by data in hepatocellular carcinoma exosomes [104]?Are the metabolic changes and increased ROS associated with the γδ17 t cells [106], indicative of alterations in the regulation of the mitochondrial melatonergic pathway?Are there interactions in the effects of melatonin and AhR agonists in the regulation of MSCs? How important is the melatonergic pathway in MSCs, including in determining mitochondrial function, exosomal content and the functioning of the exosomal mitochondria transferred to other cells?Given the AhR metabolizes melatonin and will increase the NAS/melatonin ratio, does the pre-existent state of the mitochondrial melatonergic pathway determine the consequences of AhR activation? Are the plethora of contrasting results regarding AhR activation in different cells a consequence of baseline and subsequent changes in the melatonergic pathway, including from interactions of the AhR and NF-kB?Is platelet derived serotonin utilized to synthesize NAS in the tumor microenvironment? In what cells? Does such platelet serotonin/NAS/TrkB activation not only increase survival and proliferation of cancer stem-like cells, but also increase macrophage TGFβ, thereby driving wider changes compatible with tumor survival, such as the induction of tolerogenic dendritic cells?Does tumor-derived kynurenine drive platelet AhR/CYP1B1 to increase the NAS/melatonin ratio, with the autocrine effects of NAS activating the truncated TrkB-T, leading to platelet release of pro-inflammatory cytokines and pro-angiogenic factors [92]? Is this mediated via alterations in platelet mitochondria, including ROS and ROS-driven miRNAs, leading to a distinct activated platelet phenotype?Does melatonin prevent the ‘exhausted’ γδ t cell phenotype in the tumor microenvironment, including the heightened PD-1 expression in this phenotype [107]?What regulates PEPT1/2 and OAT3 on the tumor mitochondrial membrane? Can the tumor suppress the expression of these melatonin transporters on the mitochondrial membrane?

## 5. Treatment Implications

Given the current phase 1 trials looking at the utility of AhR antagonists in the management of tumor microenvironment responses, it would be interesting to better investigate the utility of AhR inhibiting nutriceuticals, such as epigallocatechin gallate (EGCG), including intravenously. This is likely to require melatonin as an adjunctive to prevent the hepatic side-effects of relatively high-dose AhR antagonists, including those in current clinical trials.As some of the potential benefits of AhR antagonism may be mediated via a decrease in the NAS/melatonin ratio, there may be some utility in utilizing pre-existing compounds that inhibit the P2y1r and mGluR5.Given the role of platelet mitochondrial function in many ageing-associated conditions, including cancers [85], it will be interesting to determine the relevance of available SOD2 inducers/mimetics, via a targeted mitochondrial oxidant decrease, in the regulation of not only platelets, but wider cells within the tumor microenvironment. Can platelets be targeted to decrease serotonin supply to the tumor microenvironment?Would anti-CYP1B1 immunotherapy be useful across a host of diverse cancers, as some data would suggest [120,121]? Are the potential treatment implications of ongoing AhR antagonists’ phase one clinical trials mediated, at least partly, via CYP1B1 and the NAS/melatonin ratio across different cells of the tumor microenvironment?As to how the melatonergic pathways are regulated in MSCs, including by exogenous melatonin and AhR ligands may provide a basis for the utilization of these readily proliferating cells in modulating the intercellular interactions within the tumor microenvironment.

## 6. Conclusions

The mitochondrial melatonergic pathway seems an evolutionary-derived core aspect of physiological function and intercellular communication. As well as being evident in tumors, the melatonergic pathway seems evident in all cells of the tumor microenvironment, with this pathway being an important target for tumors and cancer stem-like cells in their quest to maintain survival by establishing a new intercellular homeostasis. Tumor cell release of kynurenine and its activation of the AhR is the most investigated aspect of how tumors act to regulate the mitochondrial melatonergic pathway in other cells of the tumor microenvironment. However, other targets exist including tryptophan availability and uptake, 14-3-3 isoforms and acetyl-CoA availability. The attenuation of melatonin production leads to cells in the tumor microenvironment being unable to maintain their preferred optimal function as they engage in a new homeostatic status quo. In contrast, the extracellular availability of NAS, being a BDNF mimic at TrkB, can aid the proliferation and survival of tumors and cancer stem-like cells. The wide array of dynamically regulated fluxes in the tumor microenvironment are downstream from the alterations in homeostatic regulation driven by tumor cell influence on mitochondrial function in other cells, an important aspect of which is the melatonergic pathway. Future research within this conceptualization should provide more targeted and less toxic treatments.

## Figures and Tables

**Figure 1 ijms-24-00311-f001:**
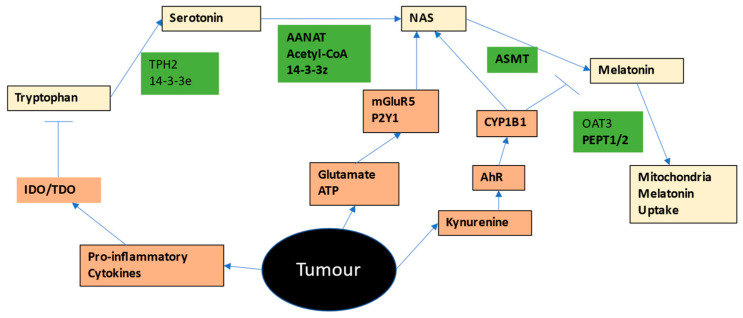
Shows the tryptophan-melatonin pathway (gold) and the factors that are required to enact this pathway (green). The tumor can release a number of factors to inhibit the tryptophan-melatonin pathway (orange), including pro-inflammatory cytokine induction of IDO and TDO, which decreases tryptophan availability. The induction of IDO/TDO in tumors leads to the production and release of kynurenine, which activates the AhR/CYP1B1, thereby metabolizing melatonin, as well as having a wide array of other effects. The direct or indirect release of glutamate or ATP will activate mGluR5 and P2Y1, which, like AhR activation, will increase the NAS/melatonin ratio. All pathway facilitating factors (green) are potential direct and indirect targets of tumors in the regulation of the tryptophan-melatonin pathway in other cells of the tumor microenvironment. Abbreviations: AANAT: aralkylamine N-acetyltransferase; AhR: aryl hydrocarbon receptor; ASMT: N-acetylserotonin O-methyltransferase; CYP: cytochrome P450; IDO: indoleamine 2,3-dioxygenase; mGluR5: metabotropic glutamate receptor 5; NAS: N-acetylserotonin; OAT: organic anion transporter; P2Y1r: purinergic receptor 2Y1; PEPT1/2: peptide transporter 1/2; TDO: tryptophan 2,3-dioxygenase; TPH: tryptophan hydroxylase.

**Figure 2 ijms-24-00311-f002:**
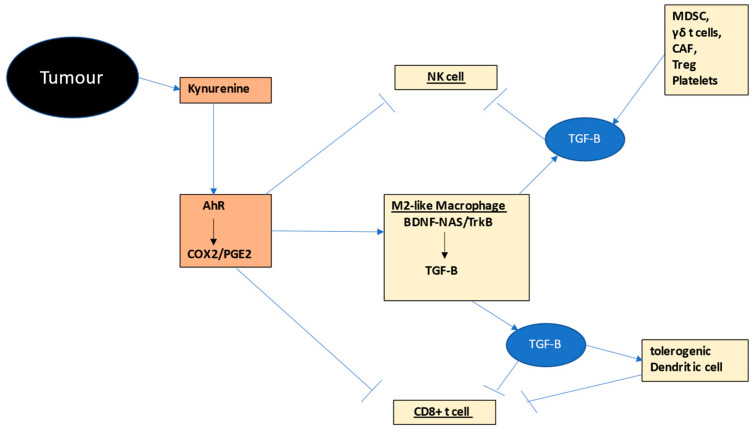
Shows how tumor-(black shade) derived kynurenine activation of the AhR drives COX2/PGE2 (bronze shade) in M2-like macrophages, NK cells and CD8+ t cells (other tumor microenvironment cells in gold shade). PGE2 in macrophages induces TrkB activation by BDNF and likely by NAS, thereby increasing TGF-B (blue shade), which then induces ‘exhaustion’ in NK cells and CD8+ t cells, as well as a tolerogenic dendritic cell, the latter further suppressing CD8+ cell activation. Numerous other tumor environment cells, including MDSCs, γδ t cells, CAF, Treg and platelets also release TGF-B. The AhR therefore interacts with other tumor fluxes, such as glutamate and ATP (Figure 1), to change the NF-kB and YY1 activation by increasing the NAS/melatonin ratio, with consequences including TGF-B release from macrophages, reinforcing NK cell and CD8+ t cell ‘exhaustion’. The NAS/melatonin ratio may therefore be an important determinant of classical processes underpinning ‘immune-checkpoint’ induction. Abbreviations: AhR: aryl hydrocarbon receptor; BDNF: brain-derived neurotrophic factor; CAF: cancer-associated fibroblasts; COX: cyclooxygenase; CYP: cytochrome P450; MDSC: myeloid-derived suppressor cell; NAS: N-acetylserotonin; NF-kB: nuclear factor kappa-light-chain-enhancer of activated B cells; NK: natural killer; PGE2: prostaglandin E2; TGF: transforming growth factor; Treg: regulator t cell; TrkB: tyrosine receptor kinase B; YY1: yin yang 1.

## Data Availability

Not applicable.

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
