# Peer review of "Tumor Microenvironment and Metabolism: Role of the Mitochondrial Melatonergic Pathway in Determining Intercellular Interactions in a New Dynamic Homeostasis"

_ijms, 2022, doi:10.3390/ijms24010311_

Round 1
Reviewer 1 Report
The manuscript provides a nice review of literature and a thorough discussion regarding our current understanding of the interplay between melatonergic pathway and the tumor microenvironment. The author has discussed the role of various cell types of the tumor microenvironment, their metabolic regulation and impact of the cross-regulation of melatonergic pathway. Though the review topic is highly relevant, and the author has done excellent work in reviewing the available literature, I would suggest a major revision due to the following issues:
1) Section 1 (Introduction) and section 2 (Tumour microenvioronment) need some more basic information and rewriting to make the reading clearer. For example, In the Introduction section, the appearance of ‘melatonergic pathway’ term looks very abrupt as there is no background information provided. Similarly, what exactly is the exhaustion of NK cells is not explained clearly. I suggest the author to merge section 3 (Tumour mitochondria as conductors of the tumour microenvironment mitochondrial orchestra) into section 1 to make the subject and objective clearer. I also suggest adding some more basic information about melatonin, its origin and mode of action in section 1. The author can also think about putting a figure about tumour microenvironment.
2) Figure 1 and Figure 2 can be improved further by using illustrators like BioRender or similar apps. Cell interaction in a visual tumour microenvironment would be easier to understand and remember.
3) There are many grammatical mistakes in the manuscript and need a careful rereading and correction by author.
4) Few words were overused or unnecessarily used in the manuscript. For example: ‘As to whether’, ‘As to’. If required, this can be replaced by ‘Whether’ and ‘about’/’from’ respectively.
5) Adding information regarding genomic regulation of melatonin regulators would also be informative.
Minor comments:
1) Rephrase the following sentences to make them easier/ clearer to understand:
a) This indicates the need for maintained OXPHOS in NK cells in order to induce a cytotoxic phenotype, with the suppression of both OXPHOS and glycolysis in NK cells being evident in cancer patients, at least partly driven by transforming growth factor (TGF)β [34] (See figure 1). As well as COX2 and NAS, the activation of the AhR increases TGFβ, adenosine A2Ar and acetyl-CoA carboxylase, the latter decreasing acetyl-CoA availability.
b) NK cells are circadian regulated, with the compromised pineal melatonin over ageing, contributing to the suboptimal mitochondrial function in NK cells occurring during the course of ‘immune ageing’, which underpins many medical conditions.
c) Factors acting to regulate the melatonergic pathway, including 14-3-3 proteins, serotonin, LAT1 and acetyl-CoA will be important to determine in neutrophils in clarifying the relevance of the melatonergic pathway in these cells, and how this is shaped by interactions within the tumour microenvironment.
d) Platelets are another cell recently associated with the tumour microenvironment, including via their interactions with circulating neutrophils [73]. The tumour microenvironment encourages the aggregation and activation of platelets, with platelets now a significant treatment target in cancer management.
e) As to whether this would then bias the availability of the mitochondrial melatonergic pathway to cells able to uptake serotonin and convert it to NAS will be interesting to determine, as this could be another means of shaping the nature of the intercellular interactions occurring within the tumour microenvironment.
f) This would also implicate other sources of tumour microenvironment NAS (and BDNF), via platelets, on angiogenesis and wider inflammatory driven processes.
g) MDSCs are also important providers of released adenosine in the tumour microenvironment, which activates the adenosine A2Ar in cytolytic cells to suppress their function, following the hydrolyzation of ATP by the ectonucleotidases, CD39 and CD73, to adenosine.
h) It is by the regulation of the mitochondrial melatonergic pathway that this is predominantly achieved.
2) Provide reference:
“OXPHOS and mTORC1 in circulating NK cells are evident in many other medical conditions”
3) “plasma membrane expression of CD62L on NK cells” : Mention how CD62L expression affects NK cell function.
4) “a major driver of the shift from an M1-like to an M2-like macrophage phenotype” : Write a sentence about how M1 and M2-like macrophage tumor response differs.
5) “However, effects may ultimately be driven by changes”: mention which effects.
6) “Interestingly, melatonin decreases the differentiation of naïve t cells into Treg, and therefore suppresses levels of TGFβ efflux”: Explain how it is relevant or connected to the previous line in the paragraph.
Author Response
Journal: IJMS (ISSN 1422-0067)
Manuscript ID: ijms-2095615
Type: Review
Title: Tumour microenvironment and metabolism: Role of the mitochondrial melatonergic pathway in determining Intercellular Interactions in a new dynamic homeostasis
Author: George Anderson
Section: Molecular Oncology
Special Issue: Mitochondrial Plasticity in Cancer
Reviewer 1:
The manuscript provides a nice review of literature and a thorough discussion regarding our current understanding of the interplay between melatonergic pathway and the tumor microenvironment. The author has discussed the role of various cell types of the tumor microenvironment, their metabolic regulation and impact of the cross-regulation of melatonergic pathway. Though the review topic is highly relevant, and the author has done excellent work in reviewing the available literature, I would suggest a major revision due to the following issues:
Response to Reviewer
Thank you for these encouraging comments.
1) Section 1 (Introduction) and section 2 (Tumour microenvioronment) need some more basic information and rewriting to make the reading clearer. For example, In the Introduction section, the appearance of ‘melatonergic pathway’ term looks very abrupt as there is no background information provided.
Response to Reviewer
Thank you for highlighting this. The following has now been added after the first mention of the melatonergic pathway in the Introduction:
“The tryptophan-melatonin pathway is shown in figure 1 (gold shading), where the enzymatic regulators and necessary co-substrates of the tryptophan-melatonergic pathway are also shown (green shading). Melatonin is classically associated with its night-time release from the pineal gland, where it is associated with the regulation of the circadian rhythm, as well as being a powerful antioxidant, anti-inflammatory, antinociceptive, antineoplastic, and optimizer of mitochondrial function. The melatonergic pathway is initiated by the stabilization of AANAT by 14-3-3zeta in the presence of acetyl-CoA. Recent data indicates the melatonergic pathway to be evident in all body cells, mostly within mitochondria. The interactions of the melatonergic pathway with mitochondria and metabolism of tumours and tumour microenvironment cells are crucial aspects of the metabolic alterations long-associated with tumours since the first publications by Warburg [3].”
Similarly, what exactly is the exhaustion of NK cells is not explained clearly.
Response to Reviewer:
Thank you for highlighting this. The following has now been added after the first mention of ‘exhaustion’: “‘Exhaustion’ refers to the failure of NK cells and CD8+ t cells to upregulate energy production (mostly glycolysis), which prevents these cells from producing a cytotoxic response, thereby preventing their capacity to kill cancer cells.”
I suggest the author to merge section 3 (Tumour mitochondria as conductors of the tumour microenvironment mitochondrial orchestra) into section 1 to make the subject and objective clearer.
Response to Reviewer:
The addition of the new paragraph in the Introduction achieves this: “The tryptophan-melatonin pathway is shown in figure 1 (gold shading), where the enzymatic regulators and necessary co-substrates of the tryptophan-melatonergic pathway are also shown (green shading). Melatonin is classically associated with its night-time release from the pineal gland, where it is associated with the regulation of the circadian rhythm, as well as being a powerful antioxidant, anti-inflammatory, antinociceptive, antineoplastic, and optimizer of mitochondrial function. The melatonergic pathway is initiated by the stabilization of AANAT by 14-3-3zeta in the presence of acetyl-CoA. Recent data indicates the melatonergic pathway to be evident in all body cells, mostly within mitochondria. The interactions of the melatonergic pathway with mitochondria and metabolism of tumours and tumour microenvironment cells are crucial aspects of the metabolic alterations long-associated with tumours since the first publications by Warburg [3].”
I also suggest adding some more basic information about melatonin, its origin and mode of action in section 1. The author can also think about putting a figure about tumour microenvironment.
Response to Reviewer
Thank you for highlighting this. The addition of the new paragraph in the Introduction achieves this: “The tryptophan-melatonin pathway is shown in figure 1 (gold shading), where the enzymatic regulators and necessary co-substrates of the tryptophan-melatonergic pathway are also shown (green shading). Melatonin is classically associated with its night-time release from the pineal gland, where it is associated with the regulation of the circadian rhythm, as well as being a powerful antioxidant, anti-inflammatory, antinociceptive, antineoplastic, and optimizer of mitochondrial function. The melatonergic pathway is initiated by the stabilization of AANAT by 14-3-3zeta in the presence of acetyl-CoA. Recent data indicates the melatonergic pathway to be evident in all body cells, mostly within mitochondria. Genomic regulation of the melatonergic pathway has been primarily investigated in the pineal gland, where the norepinephrine activation of the adrenergic Beta-1 receptor induces cAMP, which increases protein kinase (PK)A and the PKA phosphorylation of AANAT. The interactions of the melatonergic pathway with mitochondria and metabolism of tumours and tumour microenvironment cells are crucial aspects of the metabolic alterations long-associated with tumours since the first publications by Warburg [3].”
2) Figure 1 and Figure 2 can be improved further by using illustrators like BioRender or similar apps. Cell interaction in a visual tumour microenvironment would be easier to understand and remember.
Response to Reviewer: Although commercial illustrators such as BioRender can produce more aesthetic figures, I prefer the starkness of data in flow diagram format.
3) There are many grammatical mistakes in the manuscript and need a careful rereading and correction by author.
Response to Reviewer:
Apologies for this. The manuscript has now been edited for grammar.
4) Few words were overused or unnecessarily used in the manuscript. For example: ‘As to whether’, ‘As to’. If required, this can be replaced by ‘Whether’ and ‘about’/’from’ respectively.
Response to Reviewer:
Apologies for this. The manuscript has now been edited for grammar.
5) Adding information regarding genomic regulation of melatonin regulators would also be informative.
Response to Reviewer:
The following has now been added:
“Genomic regulation of the melatonergic pathway has been primarily investigated in the pineal gland, where the norepinephrine activation of the adrenergic Beta-1 receptor induces cAMP, which increases protein kinase (PK)A and the PKA phosphorylation of AANAT.”
Minor comments:
1) Rephrase the following sentences to make them easier/ clearer to understand:
- a) This indicates the need for maintained OXPHOS in NK cells in order to induce a cytotoxic phenotype, with the suppression of both OXPHOS and glycolysis in NK cells being evident in cancer patients, at least partly driven by transforming growth factor (TGF)β [34] (See figure 1). As well as COX2 and NAS, the activation of the AhR increases TGFβ, adenosine A2Ar and acetyl-CoA carboxylase, the latter decreasing acetyl-CoA availability.
Response to Reviewer
These sentences have now been changed: “The maintenance of OXPHOS is therefore required to induce an effective, cytotoxic phenotype in NK cells. The suppression of NK cell OXPHOS and glycolysis is evident in cancer patients, which is at least partly driven by transforming growth factor (TGF)β [34] (See figure 1). AhR activation not only increases COX2 and NAS, but also TGFβ, adenosine A2Ar and acetyl-CoA carboxylase, the latter decreasing acetyl-CoA availability. All of these AhR-driven changes suppress NK cell metabolism [3].”
- b) NK cells are circadian regulated, with the compromised pineal melatonin over ageing, contributing to the suboptimal mitochondrial function in NK cells occurring during the course of ‘immune ageing’, which underpins many medical conditions.
Response to Reviewer
This sentence has now been changed: “This may be important to cancer pathogenesis, given the increased risk of many cancers over the course of aging. As NK cell cytotoxicity is differentially regulated over the circadian rhythm, the attenuation of pineal melatonin production over the course of ageing, will contribute to suboptimal mitochondrial function and effective cytotoxicity in NK cells. This is likely to contribute to the course of ‘immune ageing’, which underpins many medical conditions, including cancer pathogenesis [3]. ”
- c) Factors acting to regulate the melatonergic pathway, including 14-3-3 proteins, serotonin, LAT1 and acetyl-CoA will be important to determine in neutrophils in clarifying the relevance of the melatonergic pathway in these cells, and how this is shaped by interactions within the tumour microenvironment.
Response to Reviewer
This sentence has now been changed: “Factors regulating the tryptophan-melatonin pathway, including 14-3-3 proteins and serotonin (as shown in Figure 1), will be important to determine in neutrophils. This will clarify the relevance of the neutrophil tryptophan-melatonin pathway, including how this pathway is shaped by intercellular interactions within the tumour microenvironment.”
- d) Platelets are another cell recently associated with the tumour microenvironment, including via their interactions with circulating neutrophils [73]. The tumour microenvironment encourages the aggregation and activation of platelets, with platelets now a significant treatment target in cancer management.
Response to Reviewer
These sentences have now been changed: “Recent data shows platelets to have a significant role in the tumour microenvironment, including via their interactions with circulating neutrophils [73]. The tumour microenvironment induces platelet aggregation and activation. Platelets are therefore a significant treatment target in cancer management [74].”
- e) As to whether this would then bias the availability of the mitochondrial melatonergic pathway to cells able to uptake serotonin and convert it to NAS will be interesting to determine, as this could be another means of shaping the nature of the intercellular interactions occurring within the tumour microenvironment.
Response to Reviewer
This sentence has been changed to read: “Whether platelet-provided serotonin differentially regulates the mitochondrial melatonergic pathway of tumour microenvironment cells, by having its consequences limited to cells able to uptake serotonin and convert it to NAS will be interesting to determine. Should such differential serotonin uptake and conversion to NAS occur, this would have significant impacts on the intercellular interactions within the tumour microenvironment. ”
- f) This would also implicate other sources of tumour microenvironment NAS (and BDNF), via platelets, on angiogenesis and wider inflammatory driven processes.
Response to Reviewer
This sentence has now been changed: “Platelets may therefore be significant regulators of NAS (and BDNF) production in the tumour microenvironment, with consequences for angiogenesis and wider inflammatory-driven processes.”
- g) MDSCs are also important providers of released adenosine in the tumour microenvironment, which activates the adenosine A2Ar in cytolytic cells to suppress their function, following the hydrolyzation of ATP by the ectonucleotidases, CD39 and CD73, to adenosine.
Response to Reviewer
This sentence has now been changed to read: “MDSCs release adenosine in the tumour microenvironment, thereby activating the adenosine A2Ar in cytolytic cells to suppress their function.”
- h) It is by the regulation of the mitochondrial melatonergic pathway that this is predominantly achieved.
Response to Reviewer
This has been changed to read: “This is primarily achieved via mitochondrial melatonergic pathway regulation. ”
2) Provide reference:
“OXPHOS and mTORC1 in circulating NK cells are evident in many other medical conditions”
Response to Reviewer
This sentence and references now read:
“It is important to note that such changes in NK cell function do not seem restricted to NK cell responses to cancer, as the suppression of glycolysis, OXPHOS and mTORC1 [3] in circulating NK cells are evident in many other medical conditions, including inflammatory bowel disease [38],”
3) “plasma membrane expression of CD62L on NK cells” : Mention how CD62L expression affects NK cell function.
Response to Reviewer
The following has now been added:
“Melatonin optimizes NK cell cytotoxicity [39], including elevating the ageing-associated decrease in the plasma membrane expression of CD62L on NK cells, thereby enhancing NK cell (and CD8+ t cell) ability to enter sites of tumor growth and chronic inflammation [40].”
4) “a major driver of the shift from an M1-like to an M2-like macrophage phenotype” : Write a sentence about how M1 and M2-like macrophage tumor response differs.
Response to Reviewer
The following sentence has now been added:
“The macrophage melatonergic pathway can therefore have opposing effects on M1-like and M2-like macrophages phenotypes in the tumour microenvironment, being apoptotic and pro-growth respectively [49].”
5) “However, effects may ultimately be driven by changes”: mention which effects.
Response to Reviewer
The following has now been added:
“the array of platelet effects” to read: “However, the array of platelet effects may ultimately be driven by changes in the platelet mitochondrial melatonergic pathway in association with heightened AhR activation by tumour-derived kynurenine.”
Detailing effects would be repetitive of the previous two paragraphs, given that this is the opening sentence of the following paragraph.
6) “Interestingly, melatonin decreases the differentiation of naïve t cells into Treg, and therefore suppresses levels of TGFβ efflux”: Explain how it is relevant or connected to the previous line in the paragraph.
Response to Reviewer
The following has been added:
“…. and TGFβ-induced exhaustion of cytolytic cells [105]. This highlights the importance of the local regulation of the melatonergic pathway across cell types, including Treg, in shaping patterned immune responses in the tumour microenvironment.”

Reviewer 2 Report
The manuscript entitled “Tumour microenvironment and metabolism: Role of the mitochondrial melatonergic pathway in determining Intercellular Interactions in a new dynamic homeostasis” is significant in this field of interest. This review manuscript is well-structured with enough data. However, this manuscript has minor issues needed to be addressed. Thus, I recommend this manuscript for minor revision.
Kindly check the text box border for IDO/TDO in Figure 1.
Kindly check the text format of the Figure 1 legend.
Author Response
Journal: IJMS (ISSN 1422-0067)
Manuscript ID: ijms-2095615
Type: Review
Title: Tumour microenvironment and metabolism: Role of the mitochondrial melatonergic pathway in determining Intercellular Interactions in a new dynamic homeostasis
Author: George Anderson
Section: Molecular Oncology
Special Issue: Mitochondrial Plasticity in Cancer
Reviewer 2:
The manuscript entitled “Tumour microenvironment and metabolism: Role of the mitochondrial melatonergic pathway in determining Intercellular Interactions in a new dynamic homeostasis” is significant in this field of interest. This review manuscript is well-structured with enough data. However, this manuscript has minor issues needed to be addressed. Thus, I recommend this manuscript for minor revision.
Response to Reviewer
Thank you for these encouraging comments.
- Kindly check the text box border for IDO/TDO in Figure 1.
Response to Reviewer
Thank you for spotting this. The text box border has now been standardized to the rest of the figure.
-Kindly check the text format of the Figure 1 legend.
Response to Reviewer
Thank you for spotting this. The text format has now been standardized to the rest of the manuscript.

Round 2
Reviewer 1 Report
The manuscript has been significantly improved and I do not have any further comments.